# Allosteric Coupling in Full-Length Lyn Kinase Revealed by Molecular Dynamics and Network Analysis

**DOI:** 10.3390/ijms26125835

**Published:** 2025-06-18

**Authors:** Mina Rabipour, Floyd Hassenrück, Elena Pallaske, Fernanda Röhrig, Michael Hallek, Juan Raul Alvarez-Idaboy, Oliver Kramer, Rocio Rebollido-Rios

**Affiliations:** 1Department I of Internal Medicine, Center for Integrated Oncology Aachen Bonn Cologne Düsseldorf, Faculty of Medicine and University Hospital Cologne, University of Cologne, 50931 Cologne, Germany; mina.rabipour@uk-koeln.de (M.R.); floyd@hassenrueck.de (F.H.); elenapallaske@gmail.com (E.P.); fernanda.roehrig@proton.me (F.R.); michael.hallek@uni-koeln.de (M.H.); 2Center for Molecular Medicine Cologne, 50931 Cologne, Germany; 3CECAD Center of Excellence on Cellular Stress Responses in Aging-Associated Diseases, 50931 Cologne, Germany; 4Facultad de Química, Departamento de Física y Química Teórica, Universidad Nacional Autónoma de México, Mexico City 04510, Mexico; jidaboy@unam.mx; 5Computational Intelligence Lab, Department of Computer Science, University of Oldenburg, 26129 Oldenburg, Germany; oliver.kramer@uol.de

**Keywords:** Lyn kinase, allosteric regulation, molecular dynamics simulations, cancer-associated mutations, machine learning, dynamic cross-correlation matrix, correlation network analysis, multidomain kinases, Src-family kinases

## Abstract

Lyn is a multifunctional Src-family kinase (SFK) that regulates immune signaling and has been implicated in diverse types of cancer. Unlike other SFKs, its full-length structure and regulatory dynamics remain poorly characterized. In this study, we present the first long-timescale molecular dynamics analysis of full-length Lyn, including the SH3, SH2, and SH1 domains, across wildtype, ligand-bound, and cancer-associated mutant states. Using principal component analysis, dynamic cross-correlation matrices, and network-based methods, we show that ATP binding stabilizes the kinase core and promotes interdomain coordination, while the ATP-competitive inhibitor dasatinib and specific mutations (e.g., E290K, I364N) induce conformational decoupling and weaken long-range communication. We identify integration modules and develop an interface-weighted scoring scheme to rank dynamically central residues. This analysis reveals 44 allosteric hubs spanning SH3, SH2, SH1, and interdomain regions. Finally, a random forest classifier trained on 16 MD-derived features highlights key interdomain descriptors, distinguishing functional states with an AUC of 0.98. Our results offer a dynamic and network-level framework for understanding Lyn regulation and identify potential regulatory hotspots for structure-based drug design. More broadly, our approach demonstrates the value of integrating full-length MD simulations with network and machine learning techniques to probe allosteric control in multidomain kinases.

## 1. Introduction

Protein kinases regulate diverse signaling pathways that control cellular growth, differentiation, and immune function [1,2]. Lyn kinase is a member of the Src-family kinases (SFKs) with essential functions in hematopoietic development, immune homeostasis, and cancer [3,4]. It is broadly expressed in B cells, mast cells, macrophages, platelets, and hematopoietic stem/progenitor cells, while notably absent in conventional T cells [5,6].

Lyn uniquely mediates both activating and inhibitory immune signals by phosphorylating immunoreceptor tyrosine-based activation motifs (ITAMs) and inhibitory motifs (ITIMs), thereby acting as a rheostat of immune cell activation [7]. Its dual regulatory capacity has been linked to conformational plasticity, with transitions between closed and open states shaping downstream signaling outcomes [8]. Lyn dysregulation supports proliferation and drug resistance in hematologic malignancies such as chronic lymphocytic leukemia and B cell lymphomas [9,10,11]. Emerging evidence also implicates Lyn in autoimmune and neurodegenerative disorders, as well as in solid tumor progression [7,12]. Despite its clinical relevance, Lyn remains understudied compared to other SFKs like Src or Hck, particularly in terms of its full-length conformational dynamics and allosteric regulation. This central regulatory role highlights the need for deeper structural and dynamic characterization to inform selective modulation strategies. Its dual signaling roles further complicate therapeutic targeting efforts [3,13].

Figure 1 illustrates the domain architecture of Lyn kinase and highlights functionally relevant residues, emphasizing its modular SH3–SH2–SH1 arrangement. Most structural data for Lyn are limited to the SH1 domain, either in its apo form or bound to ATP competitive inhibitors such as dasatinib [14]. Currently, no experimentally resolved structure includes all three domains, limiting our understanding of interdomain regulation and impeding the rational design of domain-specific inhibitors.

Conventional kinase inhibitors often target the conserved ATP-binding cleft, resulting in off-target effects and poor selectivity across SFKs [15]. Allosteric modulation offers a promising alternative by targeting structurally distinct residues that regulate activity through long-range coupling [7,16]. However, identifying such regulatory hotspots requires full-length, dynamics-resolved models that static crystallography alone cannot provide. In this context, mapping allosteric hubs (residues embedded in dynamic networks and often located at interdomain interfaces) can offer mechanistic insights and reveal new opportunities for selective modulation.

Advances in molecular dynamics (MD) simulations allow for the exploration of protein conformational landscapes at an atomic resolution over relevant timescales [17]. When integrated with principal component analysis (PCA), cross-correlation matrix (DCCM), and network-based methods, these approaches reveal dynamic couplings and functionally relevant communication pathways [18,19]. For multidomain kinases like Lyn, such tools are essential to link mutation, ligand binding, and long-range structural reorganization to regulatory outcomes.

In this study, we addressed these challenges by applying an integrative computational framework to characterize the full-length conformational landscape of Lyn kinase. We simulated the effects of ATP and dasatinib binding, as well as cancer-associated mutations, to investigate how these perturbations reshape domain flexibility, ATP-binding geometry, and long-range coupling. The SH3 and SH2 domains, often absent in structural studies, are explicitly modeled alongside SH1 to capture critical interdomain coordination. Using long-timescale MD simulations combined with PCA, DCCM, and residue-level network analysis, we identified dynamically embedded allosteric hubs and integration modules that govern signal propagation. A random forest (RF) classifier trained on structural and dynamic descriptors further quantifies key determinants of functional state, revealing that interdomain interactions are among the most discriminative features separating active- from inactive-like ensembles. Together, our findings establish a dynamic, network-informed framework for understanding Lyn regulation and provide an approach to context-specific modulation and structure-based targeting.

## 2. Results

To characterize the dynamic behavior of Lyn protein kinase in response to ligand binding and mutation, we performed classical MD simulations across 13 systems. These included the wildtype (WT) protein in its apo form, ATP-bound (WT-ATP), and dasatinib-bound (WT-DAS) systems, as well as five single-point mutants: K275A, E290D, E290K, I364L, and I364N, each simulated in both apo and ATP-bound conditions. All systems were simulated in three replicates for 2 μs, yielding over 78 μs of aggregate simulation time (Appendix A). Mutations were selected based on functional relevance; K275A is a known catalytic mutant [20], while E290D/K and I364L/N are naturally occurring variants identified in cancer types such as bladder urothelial carcinoma, stomach adenocarcinoma, and colorectal adenocarcinoma. Mutants I364L and I364N have been predicted to be oncogenic based on hotspot analyses in cancer genomics datasets [21].

### 2.1. Global and Local Dynamics of Lyn Protein Across Domains

We focused our analysis on global conformational stability using root mean square deviation (RMSD), local residue flexibility using root mean square fluctuation (RMSF), and mutation- or ligand-induced domain-level perturbations. Domain-resolved insights are presented for SH1, SH2, and SH3, with a focus on ATP-induced stabilization and mutation-specific dynamic signatures.

RMSD distributions of backbone atoms per domain are shown in Figure 2a, with statistical comparisons summarized in Appendix A. Among the three domains, SH3 consistently exhibited the highest deviation from the initial structure, reflecting its high mobility and functional role in regulatory switching. Unexpectedly, the WT SH3 domain displayed a bimodal distribution, suggesting potential switching between conformational states relevant to regulatory engagement. This was suppressed in SH3 mutants such as I364N, consistent with a stabilized, potentially inactive-like ensemble.

ATP-bound systems generally exhibited reduced RMSD values across all domains in Figure 2a, consistent with ligand-induced stabilization. In contrast, the WT-DAS system showed significantly higher median RMSD values in SH1 and SH2 (3.98 Å and 3.39 Å, respectively) compared to WT-ATP, indicating increased conformational variability upon dasatinib binding. This likely reflects broader target spectrum and reduced specificity of dasatinib, which may promote alternative structural rearrangements within the catalytic core. Conversely, WT-DAS displayed a significantly lower median RMSD in SH3 (7.5 Å) relative to WT-ATP, suggesting selective stabilization of this regulatory domain. These findings indicate that ATP stabilizes the kinase core, while dasatinib preferentially restricts SH3 dynamics, potentially locking Lyn into an inhibited state.

Among the mutants shown in Figure 2a, E290D and I364L exhibited higher RMSD values than E290K and I364N, respectively (particularly in the SH3 domain), indicating more substantial structural perturbations. In contrast, K275A-ATP showed moderately elevated RMSD values across all domains, consistent with disruptions of ATP binding due to loss of the conserved catalytic lysine. Statistical testing using the Wilcoxon test (Appendix A) confirmed significant differences between WT-ATP and all mutant systems, highlighting the broad impact of mutations on domain stability.

To evaluate the localized effects of mutations on residue mobility, we calculated ΔRMSF values by subtracting the RMSF of mutant systems from WT-ATP (ΔRMSF = WT − Mutant). Positive values indicate higher flexibility in the WT relative to the mutant, while negative values denote increased flexibility in the mutant. Figure 2b displays these comparisons for K275A, E290D, E290K, I364L, and I364N, with domain boundaries and residues showing prominent flexibility highlighted. Comparisons for all other systems are provided in Appendix A.

Each ΔRMSF plot spans the full 443-residue sequence, with SH3, SH2, and SH1 domains annotated. The conservative substitutions E290D and I364L displayed similar flexibility shifts with a substantial increase (negative ΔRMSF), particularly in SH1 and SH3, supporting a more permissive, active-like conformation. The more disruptive substitutions E290K and I364N exhibited pronounced reductions in flexibility across the full structure (positive ΔRMSF), consistent with an inactive-like and more compact conformation. K275A, which removes the conserved catalytic lysine, led to a marked reduction in flexibility within SH1 and adjacent C-terminal regions, reflecting structural destabilization at the catalytic core. Increased flexibility relative to the WT was observed only in a limited set of residues between positions 390 and 430.

Additionally, to improve the interpretability of mutation-induced flexibility differences in Figure 2b, residue ranges showing the largest ΔRMSF shifts were annotated. These regions include loop and linker segments across all domains (e.g., residues 60–126 in SH3, 202–210 in SH2, 281–305 and 345–369 in SH1), as well as elements of the activation loop (e.g., residues 384–407, 401–404) and distal C-terminal segments (e.g., 479–487). The observed flexibility changes at these positions suggest that they are not only structurally and functionally important but also sensitive to the effects of potential oncogenic mutations.

These ΔRMSF patterns reinforce the functional impact of each mutation, linking changes in local mobility to global effects and catalytic accessibility. Importantly, the impact of these mutations extends beyond their immediate location, perturbing interdomain linkers and allosteric communication networks. Additional comparisons in Appendix A confirmed that ATP binding generally reduces structural flexibility all over the protein structure. WT-DAS complex exhibited higher flexibility in SH1 and SH3 relative to WT-ATP, in line with previous observations.

### 2.2. Ligand and Mutation Effects on Catalytic Site Architecture

To understand how mutations influence the ability of Lyn protein to accommodate ATP, and to contrast these effects with the binding of the ATP-competitive inhibitor dasatinib, we evaluated ligand positional stability, local residue proximity, binding pocket size, binding free energy, and magnesium ion coordination. These complementary analyses offer mechanistic insights into how specific substitutions modulate active-site plasticity, ATP affinity, and conformational shifts driven by inhibitor engagement.

The ligand RMSD analysis shown in Figure 3a reveals that ATP exhibits lower mobility in all mutant systems, which displays significantly lower RMSD values compared to WT-ATP. This suggests that these mutations impose spatial constraints or alter pocket accessibility, thereby restricting ATP movement. The RMSD of dasatinib is also significantly lower than that of WT-ATP, indicating that, as a competitive inhibitor, it binds with limited conformational flexibility within the pocket and alters the native ATP binding dynamics. Statistical comparisons for all systems are summarized in Appendix A.

Figure 3b provides a comparative view of residue environments surrounding the bound ligand across the 7 holo systems. All ATP-bound systems shared a conserved core set of interacting residues, including L253, G254, V261, L263, V303, I317, the loop region around T319-A323, G325, S326, and D385. These residues delineate the canonical SH1 pocket and support stable ATP accommodation. Additional residues such as A255, A371, N372, and A384 were variably engaged in select ATP-bound mutants, indicating subtle context-dependent rearrangements. WT-ATP, E290D-ATP, and I364L-ATP share a highly similar local environment, aligning with their minimal impact on ligand RMSD. E290K, positioned within the coordination site, likely introduced charge repulsion and disrupted ${\mathrm{Mg}}^{2+}$ solvation. K275A eliminated a direct phosphate-contacting residue, diminishing ATP electrostatic stabilization.

In contrast, the WT-DAS system displayed a broader and more dispersed set of interacting residues, including V274, M294, A255, I317, and K324, many of which lie outside the core ATP-binding cleft. These observations reflect the bulkier structure of dasatinib and extended reach within the pocket, leading to altered spatial constraints. Several residues unique to WT-DAS were absent from all ATP-bound systems, further supporting a non-canonical binding mode. This divergence underscores the distinct structural consequences of inhibitor versus substrate engagement.

Figure 3c and Appendix A show that the coordination geometry and asymmetry of the magnesium ions further support these observations. In the WT-ATP system, both ${\mathrm{Mg}}^{2+}$ ions were stably coordinated in an octahedral configuration via D385, phosphate groups, and water molecules. MG1 was more tightly coordinated than MG2, as indicated by the number and proximity of surrounding waters (e.g., 2.0–2.2 Å). The oxygen atoms of D385 coordinated mainly MG1 while the triphosphate moiety from ATP was involved in the coordination of both ions. The ATP adopts a compact, horseshoe-like conformation within the binding cleft. Specific hydrogen bonds include D385–MG2 (2.0 Å), D385–K275 (2.6 Å), and E320–adenine (2.0 Å), while triphosphate interactions span 1.8–2.2 Å with coordinating residues and water molecules. This arrangement forms a robust and catalytically favorable environment stabilizing the kinase core. In mutant systems, particularly E290K and I364N, coordination was often asymmetric or incomplete. For example, E290K showed significant disruption, with D385 displaced from the vicinity of MG2 (distance > 4.0 Å), destabilizing the local coordination network (Appendix A). Such asymmetry is characteristic of kinase-inactive states where phosphate transfer is impaired. Trajectory movies illustrating dominant motions in the WT and I364N mutant, as well as ATP accommodation in the WT-ATP system, are provided as Appendix A, respectively.

Dasatinib exhibited a markedly different binding mode compared to ATP, as shown in Figure 3d. Though it occupies the ATP-binding cleft, it engages a broader and more dispersed set of contacts, particularly in β-sheets and loop regions of SH1, consistent with its promiscuous binding profile. Both the X-ray crystal structure (PDB: 2ZVA) [14] and our MD simulations show that dasatinib occupies the hydrophobic pocket formed by β-sheet and αC-helix residues, including Lys275, Met294, Val303, and Ile317. Unlike ATP, which coordinates with ${\mathrm{Mg}}^{2+}$ ions through hydrogen bonds, dasatinib engages the kinase domain primarily via π-stacking and van der Waals interactions. These include stable contacts such as M322–dasatinib (2.9 Å), A323–dasatinib (3.6 Å), and T319–dasatinib (3.3 Å), consistent with the experimental observations [14]. Although largely nonpolar, transient hydrogen bonding to hinge backbone atoms was occasionally observed. While the core binding mode is preserved an in agreement with experimental reports, the MD simulations revealed dynamic adjustments in adjacent loops and the αC helix not captured in the static crystal conformation. These flexible adaptations likely arise from the full-length context and solvent accessibility and may contribute to allosteric effects propagated beyond the ATP site.

These interactions remained highly recurrent throughout the trajectory, persisting in approximately 80% of the full simulation, highlighting their stability and relevance. Approximately 44% of the elongated dasatinib molecule extends beyond the canonical ATP pocket during the trajectory, leaving portions exposed to solvent and enabling transient interactions with adjacent surface-accessible loop regions. This occupancy pattern and interaction profile are consistent with an inactive-like conformation and help rationalize the broader allosteric effects observed in the simulations. Taken together, these findings support a robust, non-catalytic binding mode that differs mechanistically from ATP and reinforces the role of dasatinib as an ATP-competitive but conformationally selective inhibitor.

Binding free energy calculations using MM/PBSA further corroborated these findings (Appendix A). The WT-ATP system exhibited the strongest interaction ($\Delta G_{\mathrm{total}}=-28.2$ kcal/mol), consistent with high-affinity binding in a catalytically competent conformation. I364L-ATP and E290D-ATP showed moderate reductions in binding strength (−22.9 and −24.6 kcal/mol, respectively), whereas I364N-ATP and E290K-ATP displayed substantially weakened interactions (−18.1 and −17.9 kcal/mol). The K275A-ATP mutant also showed impaired affinity (−22.1 kcal/mol), in line with the disruption of catalytic lysine contacts. The WT-DAS complex yielded a binding free energy of −24.2 kcal/mol, approximately 4 kcal/mol weaker than WT-ATP, reflecting reduced interaction strength. While this shows a relatively stable complex, it is consistent with the broader and non-specific nature.

Analysis of pocket sizes via SASA measurements reinforced these trends and is shown in Figure 3e. The WT-ATP displayed the most open and catalytically accessible binding site, with an area of 2968.1 ${Å}^{2}$. In contrast, all ATP-bound mutants showed a reduction in accessible surface area. The most substantial narrowing was observed in K275A-ATP, I364N-ATP, and E290K-ATP with surface areas of 2655.6 ${Å}^{2}$, 2677.3 ${Å}^{2}$, and 2717.9 ${Å}^{2}$, respectively. E290D-ATP showed a less prominent reduction, at 2880.1 ${Å}^{2}$. I364L-ATP retained a more open binding site compared to the other mutants, with a surface area of 2900.1 ${Å}^{2}$. The WT-DAS complex presented a moderate pocket size of 2871.9 ${Å}^{2}$, smaller than WT-ATP but larger than some mutant systems, reflecting different structural effects due to inhibitor binding.

A visual comparison between WT-ATP and I364N-ATP pocket surfaces is shown in Figure 3f, highlighting the extent of structural narrowing induced by this mutation. Notably, no experimentally resolved 3D structure of full-length Lyn bound to ATP is currently available. Crystallographic data are limited to the isolated SH1 domain in complex with dasatinib [14]. Thus, these simulations uniquely characterize how ATP interacts with the full multidomain Lyn kinase and how this interaction is perturbed by mutation or inhibition.

Collectively, these results demonstrate that Lyn kinase activity depends on finely tuned spatial and electrostatic determinants at the ATP binding site. Disruption of this environment, whether through direct mutation or allosteric effects, can impair ligand retention, compromise metal coordination, and drive the kinase into an inactive-like state. Conversely, the inhibitor dasatinib induces a distinct, non-catalytic conformation characterized by increased SH3 constrains and reduced plasticity in the kinase core, likely due to its promiscuous and non-specific binding mode.

### 2.3. Dominant Global Motions Distinguish WT and Mutant Dynamics

To qualitatively assess global conformational changes across systems, we performed PCA on the trajectories of WT and apo mutants (K275A, E290D, E290K, I364L, and I364N). The analysis was based on the covariance matrix of Cα atomic fluctuations. Dominant motions were extracted from the first eigenvector (PC1), and the resulting structures represent the extreme conformations along this axis. Figure 4 illustrates the dominant motions of PC1 for each system, capturing the most relevant structural displacements sampled during the simulation time.

In the WT, PC1 captures coordinated but oppositely directed interdomain movements between SH1 and SH3, consistent with an active-like state that supports catalytic accessibility. In contrast, mutant systems exhibited altered or attenuated motion patterns. For example, K275A showed highly restricted, distorted movements suggestive of structural constraint, while E290K and I364N displayed more localized fluctuations with limited interdomain propagation. These differences imply that even single-point mutations can reshape the dominant motion landscape of the protein, potentially impairing conformational transitions necessary for allosteric regulation and enzymatic turnover. These qualitative trends motivated a more detailed investigation of inter-residue dynamic coupling via cross-correlation analysis.

### 2.4. Mutation-Induced Perturbations in Allosteric Communication Revealed by DCCMs

While PCA captures dominant directions of collective motion, it does not provide insight into how different residues move relative to one another. To assess coordinated fluctuations between residues, we computed DCCM for each system. This analysis quantifies the extent of correlated (positive values) or anti-correlated (negative values) motions between residue pairs throughout the simulations.

Figure 5 shows the DCCM extracted from the WT (a), E290K (b), and I364N (c) simulations. This analysis revealed that positively correlated motions were generally more prevalent than negative ones. In the WT, these correlations formed a highly structured and organized pattern, particularly between SH3 and SH2, as well as within flexible loop regions of SH1. Such structured correlations are characteristic of an active-like state with intact interdomain communication. In contrast, mutants displayed disrupted or fragmented correlation networks. For example, E290K exhibited localized patterns with diminished interdomain connectivity and a substantial reduction in long-range correlated motions. The SH2-SH3 axis in E290K appeared partially decoupled, indicating that the mutation imposes structural constraints that weaken dynamic communication. The I364N mutant also showed a loss of structured correlations in interdomain linker regions and an overall shift toward more confined, localized dynamics in SH3 and SH1 domains. This further supports the hypothesis that I364N impairs long-range coordination despite being located outside the catalytic site. The remaining DCCM for all other systems are provided in Appendix A for comparison.

To quantify differences in correlation behavior across systems, we calculated Pearson correlation coefficients between all pairwise DCCMs (Figure 5d). The highest similarity was observed between WT and WT-ATP (*r* = 0.95), consistent with preserved dynamic coupling upon ATP binding. Among mutants, E290K and I364L exhibited a high degree of similarity (*r* = 0.91), followed by E290D and I364L (*r* = 0.89). The correlation between E290K and I364N, as well as between E290D and I364N, was also substantial (*r* = 0.86 in both cases), suggesting a shared impact on global coordination. In contrast, the correlation between WT-ATP and WT-DAS was much lower (*r* = 0.45), highlighting distinct conformational behaviors induced by the inhibitor. Figure 5d presents a subset of systems; complete pairwise correlation data are available in Appendix A.

Together, these analyses provide a system-level view of how point mutations modulate dynamic coupling and interdomain coordination. The prevalence of positive correlations and the well-organized pattern observed in the WT contrast sharply with the fragmented correlation structures seen in mutants. Importantly, the DCCM-based correlation patterns reinforce the notion that both ligand binding and point mutations reshape the allosteric communication network of Lyn in distinct yet mechanistically coherent ways. These altered correlation landscapes serve as the foundation for subsequent network generation and community detection aimed to identify long-range communication hubs.

### 2.5. Network-Based Mapping of Allosteric Hubs

To extend our residue-level correlation analysis toward functional interpretation, we generated dynamic correlation networks and performed community detection. This approach enabled us to identify modular communication pathways and locate potential allosteric control hubs across Lyn systems. Based on structural and dynamic indicators established earlier, including domain-level RMSD/RMSF, pocket compaction, and correlation matrix integrity, WT and WT-ATP systems were classified as active-like, while all mutant systems and WT-DAS were considered inactive-like.

To evaluate how inter-residue dynamic couplings map onto the structural organization of Lyn, we applied community detection to residue–residue correlation networks. This analysis partitions the network into discrete modules of tightly correlated residues, offering a coarse-grained view of each system’s internal dynamic architecture. It is particularly valuable for multidomain proteins like Lyn, where long-range communication plays a central role in functional regulation.

To assess whether individual communities aligned with structural domains or crossed boundaries, we introduced the concept of module purity. Purity was defined as the fraction of residues within a given module that originate from its most dominant structural region. A purity near 1.0 indicates a domain-localized module, while lower values reflect cross-region mixing, a hallmark of dynamic integration. A detailed breakdown of module region composition across systems is provided in Appendix A.

Each system integration module was defined as the community with the lowest purity. This module typically includes residues from multiple structural regions and serves as a candidate hub for allosteric coordination. Most systems exhibited integration modules composed of residues from 5 structural regions. However, in the I364N-ATP system, only 4 regions were represented (SH1N, SH1C, SH2, and SH3), indicating that the interdomain linker was dynamically uncoupled from the rest of the correlated network. Moreover, across all systems, 17 out of 443 residues were not assigned to any integration module. These residues, located primarily in SH2 and SH1C, likely represent structurally flexible or dynamically isolated regions with weak network connectivity. Their consistent exclusion suggests limited involvement in coordinated domain-level dynamics (see Appendix A).

To identify residues acting as potential allosteric connectors within these integration modules, we computed betweenness centrality, which quantifies how often a residue lies on the shortest paths between other residues in the correlation network. We focused on the top 25 residues with the highest centrality per system. Although individual connector residues varied, certain residues recurred across multiple systems, suggesting a conserved role in dynamic coordination. Comparisons between apo and holo conditions of mutant systems revealed ligand-induced rewiring of connector networks, with residues gained or lost upon ATP binding often located at interdomain interfaces.

To strengthen the biological relevance of these residues, we examined the correlation between each residue centrality and system-level conformational state (active-like vs. inactive-like). Residues with high positive or negative correlations were likely critical for distinguishing functional states. To further refine the list, we computed a consistency score reflecting recurrence across a range of correlation thresholds and a mixing score quantifying the compositional heterogeneity of the integration module. These metrics were combined into a unified score.

Finally, we integrated structural interface information derived from MD-based contact analysis. Residues were annotated based on their presence at SH1, SH2, SH3, or linker interfaces across systems. The final selection was based on an interface-weighted score that combined dynamic centrality with structural positioning. Residues with a score exceeding 0.5 were defined as allosteric hubs (n=44), spanning all domains, though fewer were found in the SH2. Many of these hubs localize to the SH2–SH1 linker, SH1N lobe, and SH3-SH1C boundary regions well-positioned to mediate long-range dynamic coupling. Scores for all 443 residues are provided in Appendix A; see Methods for a full description of the scoring procedure.

The domain-level organization of connector residues is depicted in Figure 6a, highlighting their distribution across SH3, SH2, SH1N, and SH1C domains, and the interdomain linker. Rather than remaining confined within individual regions, these residues form cross-domain clusters. Dashed connections between modules indicate putative interdomain communication paths derived from the correlation network. When projected onto the 3D structure (Figure 6b), these residues form a spatially distributed network spanning the entire protein architecture. This organization supports the interpretation that these positions constitute an embedded allosteric wiring system capable of mediating long-range coupling and responding to both mutation and ligand-induced perturbations.

The allosteric hubs identified here were most prominent in the active systems (WT and WT-ATP) than in mutants and WT-DAS. This suggests that dasatinib binding, like disruptive mutations, destabilizes the native allosteric network rather than shifting it to an alternative configuration. The loss of key connector residues across some inactive-like states underscores a breakdown in long-range coordination and supports the distinction between catalytically competent and inhibited conformations.

### 2.6. Discriminative Structural Features Define Domain-Level Determinants of Functional States

Given the altered correlation networks and reduced connectivity of key allosteric hubs in mutant systems, we next assessed whether structural and dynamic features extracted from MD trajectories could systematically distinguish different functional states. To this end, we trained a RF classifier using 16 interpretable features derived from each simulation snapshot. These descriptors, listed in Appendix A, were selected based on prior observations and included residue–residue distances central to ATP coordination and domain cross-talk (e.g., K275–E290, D385–K275), activation loop geometry (e.g., R-spine angle), and interdomain interactions (e.g., SH2–SH3, SH3–SH1N).

The dataset included approximately 78,000 frames, each represented by these 16 features. Systems were labeled as “active-like” (WT, WT-ATP) or “inactive-like” (WT-DAS, all mutants) based on structural, dynamic, and correlation-based criteria established earlier. Importantly, while the classifier was not primarily developed for predictive application, it provided a means to evaluate which features best encode conformational differences across the ensemble. The classifier achieved, in the test dataset, an overall accuracy of 96%, with high recall and precision for both active-like and inactive-like classes. All metrics used to evaluate performance are summarized in Table 1. The area under the ROC curve (AUC) was 0.98, as shown in Figure 7a.

Feature importance rankings are shown in Figure 7b. The most important features included the K275–E290 salt bridge, SH3–SH1N distance, and E290–D385 coupling interactions that span both regulatory and catalytic domains. Additional high-ranking descriptors such as SH2–SH3 separation, D385–K275 distance, and the R-spine angle further emphasize the importance of interdomain coordination. The top seven features together accounted for approximately 66.7% of the total permutation importance, underscoring their dominant contribution to functional classification.

This analysis supports the notion that interdomain interactions are among the most informative determinants of functional state in full-length Lyn kinase. The classifier reinforces and quantifies insights gained from structural, dynamical, and network-based analyses, validating that conformational coupling between SH3, SH2, and SH1 domains plays a defining role in distinguishing active-like and inactive-like ensembles.

## 3. Discussion

Protein kinases function as dynamic molecular switches whose regulatory mechanisms rely on conformational flexibility and interdomain communication [22]. Among SFKs, Lyn is a critical player in hematopoietic signaling, immune regulation, and cancer progression [3,23]. However, its full-length structure remains experimentally unresolved, and mechanistic insights have primarily focused on the SH1 domain [14,24]. This study presents, to our knowledge, the first long-timescale MD analysis of full-length human Lyn kinase, including SH3, SH2, and SH1 domains, across apo, ATP-bound, and dasatinib-bound states in both WT and mutant forms.

Our results show that ATP binding stabilizes an active-like conformation by reinforcing SH1 integrity and maintaining interdomain coordination. In contrast, dasatinib, an ATP-competitive inhibitor, induces a distinct structural response characterized by increased SH1 and SH2 flexibility and reduced SH3 mobility. Dasatinib binds predominantly through π-stacking and van der Waals interactions, occupying the hydrophobic pocket near the hinge and β-sheet/αC-helix regions consistent with the SH1 domain structure reported by Williams et al. [14]. This redistribution resembles previous observations in Src [25,26] but extends them to Lyn in a multidomain context. These differences were further supported by binding free energy calculations: ATP showed the strongest affinity (−28.2 kcal/mol), whereas dasatinib exhibited moderately reduced affinity (−24.2 kcal/mol), consistent with its broader, non-specific binding mode. These results reinforce the notion that ATP stabilizes a high-affinity, catalytically competent state, while both mutation and inhibitor binding attenuate this stability through distinct structural mechanisms.

Cancer-associated mutations, notably E290K and I364N, had distinct allosteric effects. E290K disrupted ${\mathrm{Mg}}^{2+}$ coordination, weakening phosphate handling and impairing catalysis [27]. I364N, located distal to the active site, decoupled SH3 from the linker region, leading to broader conformational shifts. I364N has not previously been characterized structurally, but its predicted oncogenic potential [21] aligns with our finding that it impairs long-range coordination.

The community detection analysis performed on DCCM revealed that interdomain integration modules, spanning SH3, SH2, and SH1, are enriched in WT and ATP-bound states but are disrupted by mutations and inhibitor binding. These modules harbor 44 allosteric hubs, several of which overlap with hinge-adjacent regions and known regulatory motifs. Their reduced representation in mutant systems suggests that mutations attenuate conserved long-range communication rather than rewire it.

Sixteen descriptors used to train the RF classifier highlight essential biophysical determinants underpinning Lyn kinase regulation. Salt bridges involving E290-K275 and D385-K275 directly affect ATP-binding pocket stability and catalytic readiness, highlighting their central roles in kinase activation dynamics. Distances between SH3 and SH1N domains, and SH2 and SH1C domains, underscore interdomain communication critical for transmitting allosteric signals that toggle kinase states. The integrity of the R-spine further emerged as an essential determinant of kinase conformation, reflecting its conserved role in sustaining active states across kinases. Residue pairs such as Y397-R366 and Y397-E289 offer insights into activation-loop phosphorylation events, fundamental to the catalytic function. Additionally, interactions like K275-M294 reflect localized conformational adjustments essential for nucleotide coordination. Furthermore, domain-center distances such as SH3–SH2 and SH1–Lobes are indicative of larger-scale conformational rearrangements, essential for understanding the global regulatory mechanisms underpinning kinase autoinhibition and activation. Several distal interactions (I77-N449, D498-K137, R390-K361) also surfaced as discriminative, emphasizing the role of long-range intramolecular coupling in kinase structural dynamics. Such long-distance correlations reinforce the concept of allosteric networks extending beyond immediate catalytic regions, linking peripheral structural elements to the core functional domains.

Primarily designed as an interpretability tool, the RF classifier identifies structural descriptors that most effectively distinguish active-like from inactive-like ensembles, rather than aiming to optimize predictive performance. This approach enables the systematic identification of domain-spanning mechanisms and structurally informative features. The top seven features accounted for approximately 67% of total feature importance and were consistent with both DCCM-derived integration hubs and ΔRMSF results, demonstrating multi-method convergence. Moreover, because many of these structural elements (especially R-spine, hinge salt-bridge networks, and domain-lobe couplings) are conserved across SFKs and other kinases, our findings likely generalize beyond Lyn. Such conserved architecture provides a framework for investigating allosteric control in related kinase families. However, their precise role and impact are context-dependent, reflecting evolutionary adaptations and functional specialization within kinase families. This interpretable approach complements recent artificial intelligence (AI)-based tools for kinase conformational prediction [28,29], offering physics-based granularity and mechanistic resolution.

Comparison with other SFKs highlights both shared and unique features. While Src and Hck also exhibit ATP-dependent stabilization [30], most prior studies are confined to SH1-only models [31,32]. Allosteric regulation has been explored within the SH1 domain in Src using computational and experimental approaches [33,34], but full-length SFK simulations are limited to a few examples [35,36,37], and our study extends this space by revealing Lyn-specific dynamics. In particular, SH3–SH1N interactions appear to play a more central role in active-state stabilization in Lyn than in Src or Hck, where SH2–SH3 coupling dominates [38]. This finding aligns with structural data from Hck [39] and a recent conformational analysis of Src [40], indicating that SFK activation is modular and subtype-specific. Moreover, these observations support the idea of alternative active conformations within individual Src-family members with distinct signaling properties [41,42].

From a therapeutic perspective, these insights support the rational development of context-specific Lyn modulators. Inhibitor resistance and signaling plasticity in hematologic malignancies underscore the need for strategies beyond ATP-site targeting [4]. The allosteric modulation of kinases represents a promising therapeutic strategy, enabling selective inhibition via binding to regulatory sites outside the ATP cleft. Our results indicate that targeting allosteric hubs, particularly those mediating SH3–SH1N communication and R-spine integrity, may yield improved selectivity and functional control. These insights could support the development of more selective allosteric modulators [11,43,44].

This study opens promising avenues at the interface of MD and rational drug design. Long-timescale MD simulations can reveal cryptic pockets and conformational transitions beyond static crystal structures. Combined with network theory analysis and ML-based feature selection, these tools enable the systematic identification of allosteric hotspots and structural vulnerabilities. Data-driven models are already emerging to predict allosteric sites from protein dynamics and can be trained to associate conformational signatures with functional outcomes. Several allosteric hubs identified here may serve as candidates for experimental validation through targeted mutagenesis or conformational biosensors such as FRET or HDX-MS [45], providing a path toward structural verification. Ultimately, this computational framework, merging physics-based MD, graph analysis, and AI, holds significant promise for the next generation of kinase drug discovery. Our integrative strategy, demonstrated here for Lyn, offers a transferable blueprint for dissecting kinase regulation and informing allosteric drug design across the broader kinome.

## 4. Materials and Methods

### 4.1. Homology Modeling of Full-Length Lyn Protein

A full-length structural model of Lyn kinase (residues 60 to 502) was generated using Modeller [46]. High-resolution crystal structures of homologous SFKs were used as templates: 6NMW (human Lyn SH3 domain), 4TZI (mouse Lyn SH2 domain), and 5H0B (human full-length Hck). Model quality was assessed using the ERRAT server [47], yielding a global quality factor of 99.1, indicative of a high-confidence model suitable for molecular simulations. The final model was energy-minimized using the steepest descent algorithm in GROMACS 2022.04 [48].

### 4.2. Mutant Generation and Small Molecule Incorporation

Single-point mutations at residues I364 (I364L and I364N), E290 (E290D and E290K), and K275 (K275A) were introduced into the full-length Lyn model using the SCWRL4 program with default parameters [49]. These variants were selected based on their presence in cancer genomics datasets or previously reported functional relevance. For ATP-bound systems, one ATP molecule and two ${\mathrm{Mg}}^{2+}$ ions were positioned in the catalytic cleft by structural alignment to the kinase complex in PDB ID: 1ATP. For the system containing the ATP-competitive inhibitor dasatinib, coordinates were extracted from the crystal structure (PDB ID: 2ZVA), superimposed onto the modeled apo Lyn structure, and energy-minimized to resolve steric clashes and refine the binding pose. The chemical structure of dasatinib was prepared using ChemDraw (https://www.perkinelmer.com/category/chemdraw, accessed on 15 June 2025) and is provided as Appendix A. The molecule has a molecular formula of C_22_H_26_ClN_7_O_2_S and includes a thiazole core and substituted aniline moiety, structural features that enable π-stacking and van der Waals interactions within the ATP-binding pocket of kinases. Both WT and mutant models, either unbound (apo), ATP-bound (holo), or dasatinib-bound (holo), were used as initial structures for MD Simulations.

### 4.3. Quantum Mechanical Calculations

Geometry optimization, frequency calculations, and population analyses of dasatinib were performed with the Gaussian 16 package of programs [50] using the B3LYP functional and the 6–31 G(d) basis set. Geometry optimization and frequency calculations were carried out in solution, using the SMD continuum model and water as solvent. SMD is considered a universal solvation model due to its applicability to any charged or uncharged solute in any solvent or liquid medium [51]. Vibrational analysis indicates that geometries correspond to minima. Computed electrostatic potential (ESP)-derived atomic charges were used later for MD simulations.

### 4.4. Classical MD Simulations

A summary of all simulations is presented in Appendix A. We conducted 78 µs of MDs (3 replicates per system, to ensure reproducibility) using the all-atom additive CHARMM36 force field [52] with optimized magnesium binding parameters [53] in GROMACS 2022.04 [48]. The protein was placed in the center of a rhombic dodecahedron box with a minimal distance from the structure to the box boundaries of 10 Å. The TIP3 explicit water model [54] was used to solvate the system. Sodium and chloride ions were added to neutralize the systems at an ionic concentration of 0.15 mol L^−1^. Equilibration included 5 ns in the NVT ensemble with restrained heavy atoms, followed by 5 ns in the NPT ensemble without restraints. Temperature was stabilized at 310 K using V-rescale thermostat [55] and pressure at 1 atm by the Parinello–Rahman barostat [56], respectively. Electrostatic interactions were calculated using the Particle Mesh Ewald [57], and LINCS [58] was used for bond constraints. Production MD simulations were computed for 2 µs each on a GPU (Nvidia GeForce RTX 4090, Cuda 12.2. Koeln, Germany) with a 2 fs time step. Coordinates, velocities, and energies were saved every 1 ns.

### 4.5. Trajectory Analyses

Trajectory analyses were performed using GROMACS tools to assess structural stability and residue-level fluctuations. Backbone RMSD was calculated on backbone atoms using gmx rms, and RMSF was calculated on Cα atoms using gmx rmsf. Distances and angles were computed using gmx distance and gmx angle, respectively, and all outputs were processed and analyzed in R Statistical Software (v4.4.1; R Core Team 2021 [59]). Representative structures were extracted from the trajectories by clustering frames based on mutual backbone RMSDs using the PAM algorithm implemented in the cluster R package v2.1.8.1 [60]. Clustering quality was evaluated using silhouette coefficients computed with the fpc R package v2.2-13 [61]. For each system, the most populated cluster was selected, and its medoid, the frame with the lowest average RMSD to all other frames within the cluster, was defined as the representative structure.

### 4.6. Interface Residues

Interdomain interface residues were identified using the InterfaceResidues plugin in PyMOL. v3.1.0 This method estimates solvent-accessible surface area (SASA) changes to detect residues involved in domain–domain interactions. The algorithm first calculates the SASA of the full complex; then, it separates the complex into two regions (e.g., individual domains or chains) and computes the SASA for each independently. The difference between the summed individual SASAs and the complex SASA (ΔSASA) reflects the buried surface area upon domain association. Residues exhibiting a SASA reduction greater than 1.0 ${Å}^{2}$ were classified as interface residues.

### 4.7. Binding Pocket Surface Area Estimation

To estimate binding pocket size across systems, the SASA of the ligand (ATP or dasatinib) was calculated. Pocket residues were predefined as all residues within a 5 Å radius of the ligand, effectively delineating the binding site based on spatial proximity. The SASA of this ligand-surrounding region was then computed using GROMACS 2022.04 analysis tools and used as an approximation of pocket surface area.

### 4.8. Projection of Dominant Motions Along Principal Components

To investigate large-scale conformational dynamics, we projected atomic displacements along the principal components derived from each system trajectory. PCA was performed in GROMACS using the positional fluctuations of Cα atoms. The covariance matrix was calculated with gmx covar, and eigenvectors were analyzed using gmx anaeig. To visualize the dominant motions, extreme projections along the first principal component (PC1) were generated, representing the maximum conformational deviations along this mode. A series of interpolated structures were then constructed to illustrate the transition between these extremes. These structures were visualized using VMD [62], enabling the interpretation of system-specific differences.

### 4.9. Dynamical Cross-Correlation Analysis

Dynamical cross-correlation analysis was performed to identify coupled motions within the protein. Pairwise cross-correlation coefficients of atomic fluctuations were computed using the dccm function from the Bio3D package [63]. Cα atoms were used to align trajectory frames, and an N×N cross-correlation matrix was generated, where *N* is the number of residues. Each matrix element reflects the degree of dynamic correlation between atomic displacements. The results were visualized as dynamical cross-correlation matrices.

### 4.10. Community Detection and Integration Module Identification

Community detection was performed using the Louvain algorithm on DCCM to examine how residue–residue correlations organize into structural communities. For each system, an undirected, weighted graph was constructed from the upper triangle of the dynamical cross-correlation matrix, retaining edges with absolute correlation values above a threshold of 0.8. Residues were represented as nodes, and edge weights reflected the magnitude of dynamic correlation. Communities (modules) were identified based on graph topology using the Louvain algorithm, and module memberships were mapped to known structural regions (SH1N, SH1C, SH2, SH3, and the interdomain linker). To assess the degree of structural integration, the purity of each module was calculated as the fraction of residues belonging to the dominant structural region. Modules with low purity, spanning multiple domains, were considered “integration modules”. The lowest-purity module from each system was selected for downstream analysis.

### 4.11. Identification of Allosteric Hubs from Networks and Interface Features

Residues with potential allosteric significance were identified based on their betweenness centrality within each system’s integration module. Betweenness values were computed on the full residue–residue correlation network, and residues were ranked according to their centrality within each system. To capture functionally relevant variation, the correlation between each residue’s centrality profile and the binary classification of systems as active-like or inactive-like was calculated. For a given residue *r*, the vector of betweenness values across all *N* systems was denoted as $b_{r}=\left(b_{r,1},\ldots ,b_{r,N}\right)$, and the corresponding activity labels were encoded as $y=(y_{1},\ldots ,y_{N})$, where $y_{s}=1$ for active-like systems and $y_{s}=0$, otherwise. The Pearson correlation $\rho_{r}=\mathrm{corr}\left(b_{r},y\right)$ was used to quantify the relationship between centrality and conformational state. To further characterize residue behavior, a module-mixing index $M_{r,s}$ was calculated for each residue *r* in each system *s*, reflecting the compositional heterogeneity of its integration module. The mean module-mixing score across systems was computed as follows:(1)$$\mu_{r}=\frac{1}{N}\sum_{s=1}^{N}M_{r,s}.$$

These two quantities were combined to produce a mixing-weighted correlation score, which integrates functional sensitivity and modular diversity:(2)$$\mathrm{mix}\_\mathrm{weighted}\_{\mathrm{corr}}_{r}=\rho_{r}\;\cdot \;\mu_{r}.$$

To incorporate structural context, domain interface frequency was introduced as a weighting factor. The criteria for identifying whether a residue belongs to an interdomain interface are detailed in Section 4.6. For each residue *r*, the number of systems in which it was part of an interdomain interface was recorded as $k_{r}$ and normalized as $f_{r}=\frac{k_{r}}{N}$. This value was used to compute an interface-weighted score:(3)$$\mathrm{interface}\_\mathrm{weighted}\_{\mathrm{score}}_{r}=\mathrm{mix}\_\mathrm{weighted}\_{\mathrm{corr}}_{r}\;\cdot \;\left(1+f_{r}\right),$$
which reflects both dynamic and interface-associated significance. Residues with an interface-weighted score greater than 0.5 were classified as allosteric hubs. The full set of scoring results is provided in Appendix A.

### 4.12. Random Forest Classification

A supervised classification pipeline was implemented using the scikit-learn library to evaluate which structural and dynamic features most effectively distinguish active-like from inactive-like conformational ensembles of Lyn kinase. Sixteen interpretable features were extracted from MD trajectories (Appendix A).

Approximately 78,000 simulation frames were analyzed, each labeled as “active-like” (WT, WT-ATP) or “inactive-like” (WT-DAS and all mutant systems). All features were treated as continuous variables and scaled using Min–Max normalization. The dataset was randomly shuffled and partitioned into training (80%) and testing (20%) sets using stratified sampling to maintain class distribution. To address class imbalance, a modeling pipeline combining the Synthetic Minority Oversampling Technique (SMOTE) [64] with a RF classifier was applied. Hyperparameters including the SMOTE sampling ratio, number of estimators, maximum tree depth, and class weighting were optimized via grid search with five-fold cross-validation.

The final model was trained using the following optimal parameters: n_estimators = 1000, max_depth = None, class_weight = None, and smote__sampling_strategy = 0.5. Model performance was evaluated on the test set using standard classification metrics, including accuracy, precision, recall, F1-score, and the AUC. Feature importance was computed using both Gini index and permutation methods and is reported with respect to the original feature names.

### 4.13. Statistics

Statistical significance was assessed using the Wilcoxon signed-rank test (two-sided, significance threshold p<0.05). Detailed numerical data supporting the analyses are provided in the Appendix A.

## 5. Conclusions

This work provides a comprehensive structural and dynamical characterization of how ligand binding and cancer-associated mutations modulate the conformational ensemble, interdomain communication, and allosteric architecture of full-length Lyn kinase. Through long-timescale MD simulations, dynamic residue network analysis, and machine learning-based classification, we demonstrate that ATP and dasatinib stabilize distinct functional states, and that specific mutations, particularly I364N and E290K, disrupt key allosteric couplings and shift the kinase toward inactive-like conformations. By analyzing the entire SH3–SH2–SH1 assembly, we identified a set of dynamically embedded residues that serve as putative allosteric hubs and define structural features that robustly differentiate active from inactive ensembles. These insights deepen our understanding of multidomain Lyn regulation and highlight the value of full-length, dynamics-driven approaches in supporting structure-based drug discovery and mutation-specific therapeutic strategies.

## Figures and Tables

**Figure 1 ijms-26-05835-f001:**
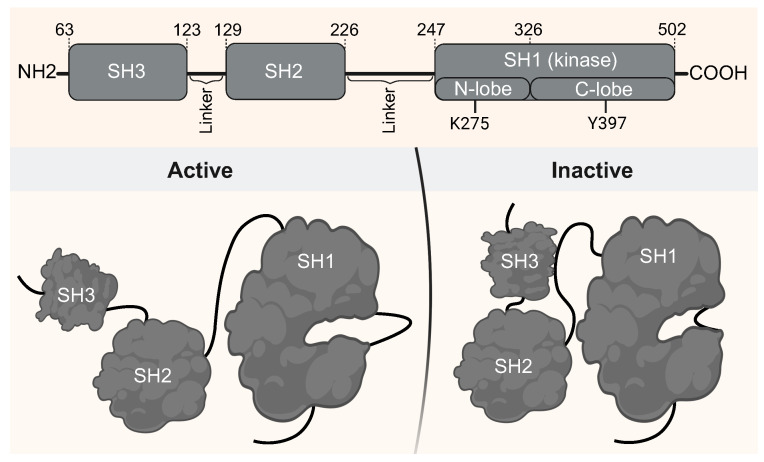
Domain organization and conformational states of full-length Lyn kinase. **Top**: Schematic representation of Lyn modular architecture, comprising SH3, SH2, and SH1 (kinase) domains, connected by flexible linkers. Key catalytic residues (K275 in the N-lobe and Y397 in the C-lobe) are indicated. **Bottom**: Cartoon depiction of the conformational switch between the active (**left**) and inactive (**right**) states. In the inactive conformation, SH3 engages the SH2–SH1 linker and SH2 interacts with the phosphorylated C-terminal tail, stabilizing autoinhibition. In the active state, SH3 is displaced, allowing for an open, catalytically competent conformation. Figure created in BioRender. Rebollido-Rios, R. (2025), https://BioRender.com/w26lvf0.

**Figure 2 ijms-26-05835-f002:**
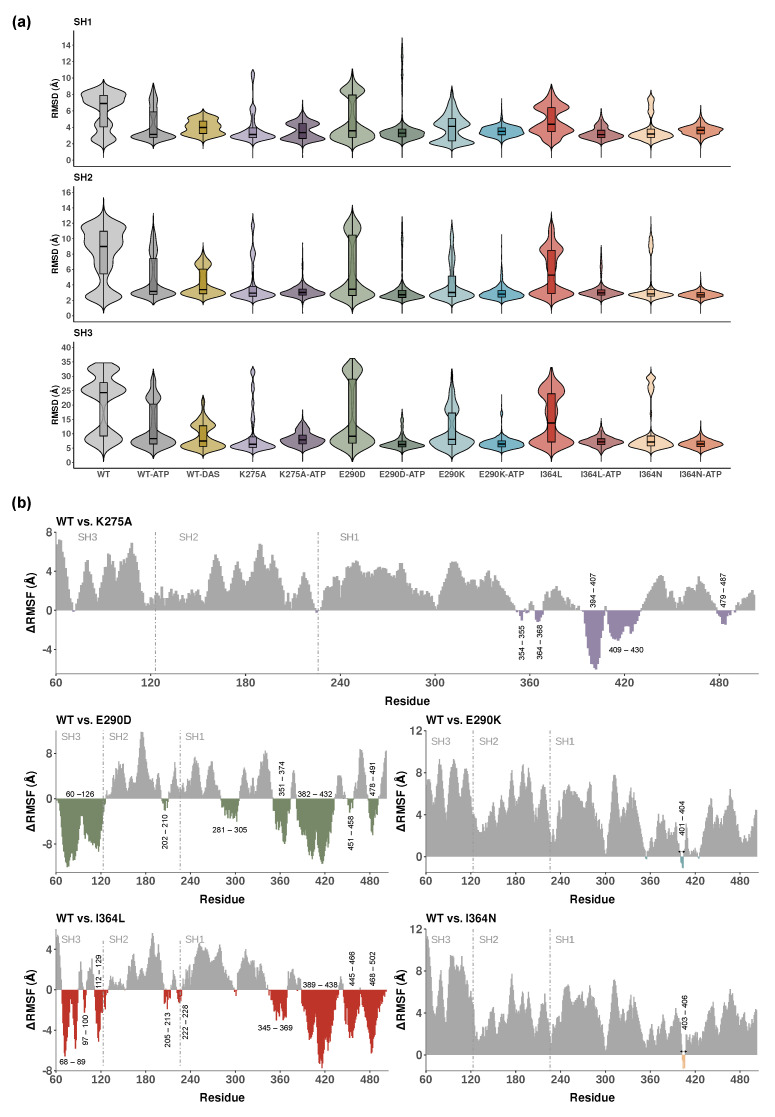
Domain-resolved analysis of structural deviations and local flexibility across Lyn kinase systems. (**a**) Violin plots showing backbone RMSD distributions per domain (SH1, SH2, SH3) for all 13 systems. RMSD values reflect deviations from the starting structure and indicate domain-level stability. (**b**) ΔRMSF plots for key mutants (K275A, E290D, E290K, I364L, I364N), calculated as the difference in RMSF of the WT relative to the mutant (WT − Mutant). Positive values indicate reduced flexibility in mutants, while negative values denote increased flexibility. Residues showing the most prominent flexibility shifts are explicitly labeled, and domain boundaries (SH3, SH2, SH1) are annotated and separated by dashed vertical lines.

**Figure 3 ijms-26-05835-f003:**
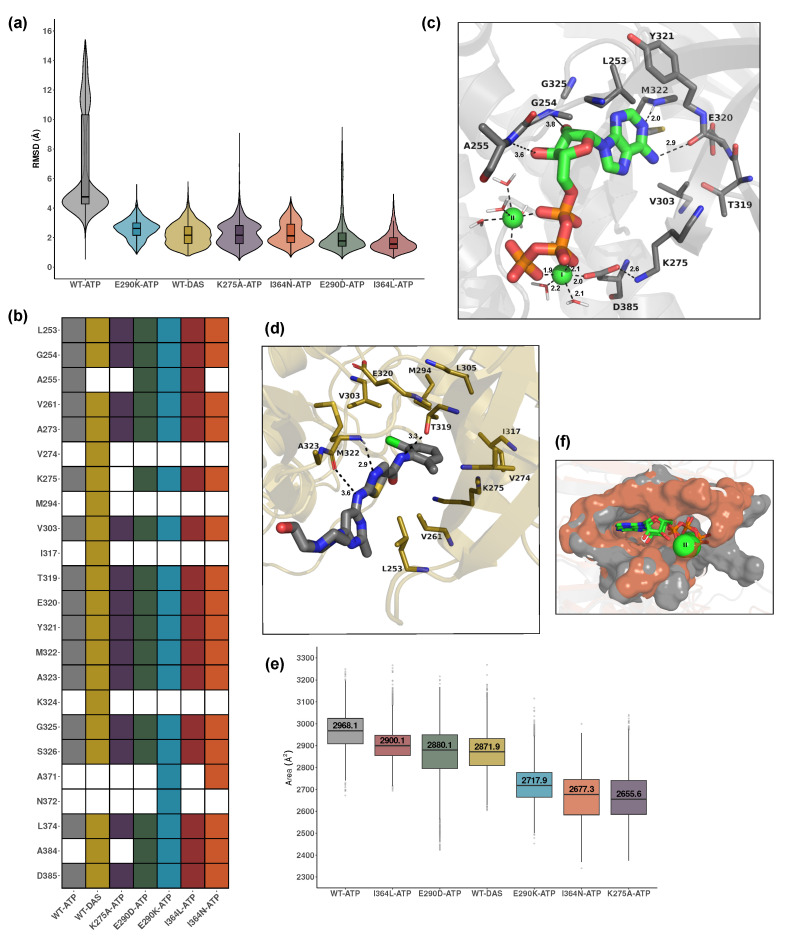
Structural and dynamic insights into ATP and dasatinib binding. (**a**) Violin plots showing RMSD distributions of ATP and dasatinib ligands across all holo systems, indicating ligand positional stability within the binding site. (**b**) Heatmap of residues within 5 Åof the ligand in each system, reflecting the composition of the local binding environment. Shared and unique contacts across systems illustrate ligand-specific interaction patterns. (**c**) Representative binding of ATP in the WT-ATP system, highlighting key coordinating residues and Mg^2+^-mediated contacts. Dashed lines indicate coordination distances between ATP, residues, and metal ions. Interacting residues are colored as: carbon in gray, nitrogen in blue, oxygen in red. Mg^2+^ ions are shown as green spheres. (**d**) Representative dasatinib binding in WT-DAS, showing broader and more dispersed contact residues including loop and β-sheet regions. Dashed lines indicate key intermolecular distances involved in the binding. Interacting residues are colored as: carbon in mustard, nitrogen in blue, oxygen in red. (**e**) Pocket size estimated based on the SASA of the ligand-defined region in each system, reflecting mutation- or inhibitor-induced compaction. (**f**) Structural comparison of pocket accessibility between WT-ATP and I364N-ATP systems, highlighting the narrowing of the cavity in the mutant.

**Figure 4 ijms-26-05835-f004:**
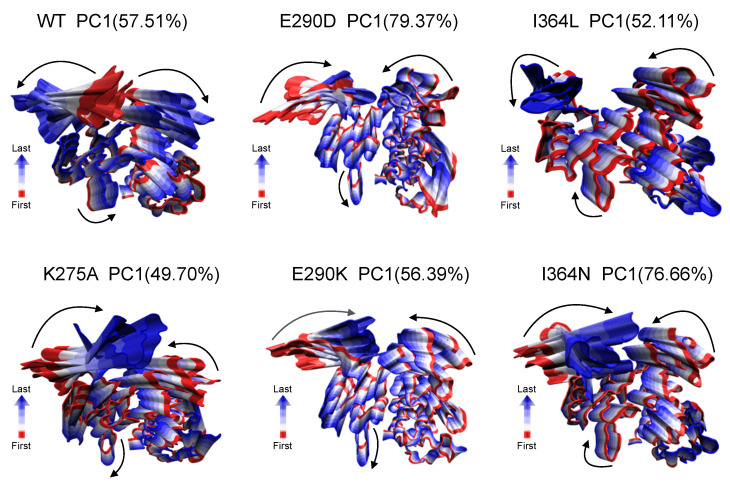
Dominant global motions captured by principal component analysis (PCA). Snapshots representing extreme projections along PC1 are shown for WT and mutant systems (K275A, E290D, E290K, I364L, and I364N). PC1 accounts for the highest variance in atomic fluctuations and captures large-scale domain movements. Structures are colored from red (start of motion) to blue (end of motion), illustrating the directionality and amplitude of motion. Arrows indicate the dominant direction of displacement along PC1 for SH3, SH2, and SH1 domains. Percentage values in parentheses denote the variance explained by PC1 in each system.

**Figure 5 ijms-26-05835-f005:**
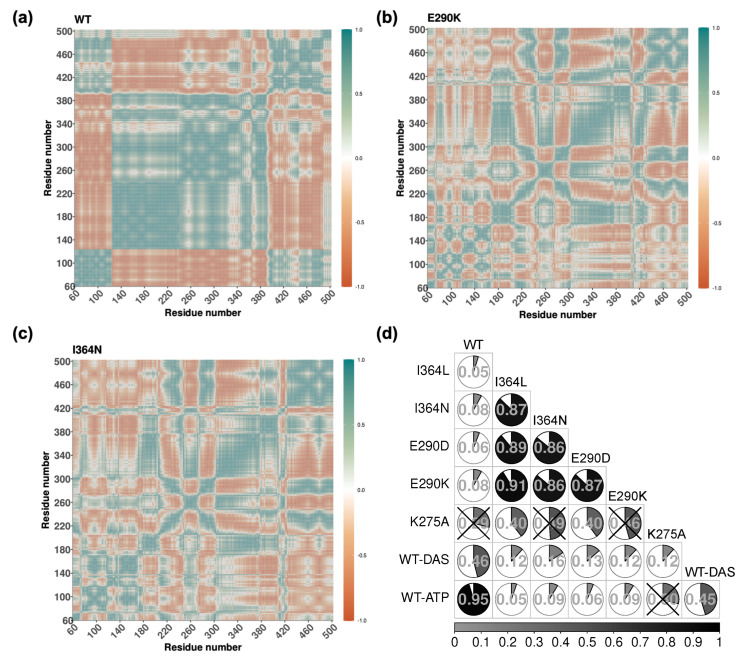
(**a**–**c**) DCCMs for the WT, E290K, and I364N systems showing pairwise correlations of Cα atom displacements. Positive correlations are shown in teal, negative in red, with intensity reflecting correlation magnitude. The WT exhibits well-structured long-range correlations across domains, indicative of coordinated dynamics. Mutants E290K and I364N display fragmented and localized correlation patterns, reflecting impaired interdomain communication. (**d**) Pearson correlation coefficients between DCCMs across all systems. Each cell is represented as a pie chart with shading proportional to the correlation value displayed at the center. Cells marked with “X” denote comparisons where the correlation was not statistically significant.

**Figure 6 ijms-26-05835-f006:**
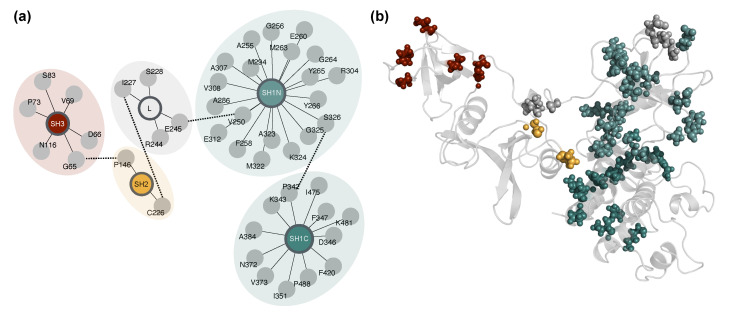
Spatial and network organization of the 44 allosteric hubs identified across systems. (**a**) Network-based schematic showing the distribution of allosteric hub residues across Lyn kinase domains. Hubs are grouped and colored by structural regions (SH3, SH2, interdomain linker [L], SH1N, SH1C). Dashed edges indicate putative interdomain communication paths inferred from a correlation network topology. (**b**) Three-dimensional spatial mapping of the same 44 residues onto the WT full-length Lyn structure. Hub residues are shown as spheres and colored as in (**a**), highlighting their broad distribution and role in bridging multiple domains. This architecture supports a model of embedded allosteric wiring responsive to ligand binding and mutation.

**Figure 7 ijms-26-05835-f007:**
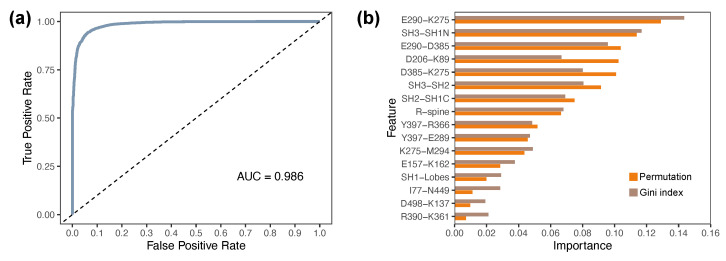
Random forest classifier distinguishes active- and inactive-like Lyn ensembles based on structural and dynamic features. (**a**) Receiver operating characteristic (ROC) curve showing classifier performance, with an AUC of 0.986, indicating excellent discrimination between functional states. (**b**) Feature importance analysis based on Gini index and the permutation method. Top-ranking descriptors include the E290–K275 and E290–D385 distances, SH3–SH1N separation, and D385–K275 interaction.

**Table 1 ijms-26-05835-t001:** Metrics used to evaluate the performance of the RF classifier.

	Precision	Recall	F1-Score
Active-like	0.91	0.84	0.87
Inactive-like	0.97	0.98	0.98
Accuracy		0.96	
Macro average	0.94	0.91	0.92
Weighted average	0.96	0.96	0.96

## Data Availability

The initial full-length Lyn kinase models, along with input structures and representative output coordinates from all production simulations (in PDB format), are available in the following GitHub repository: https://github.com/RebollidoRiosLab/LynFull_AllostericNetwork.git. Additional simulation data and analysis materials are available from the corresponding author upon reasonable request.

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
