# Peer review of "Allosteric Coupling in Full-Length Lyn Kinase Revealed by Molecular Dynamics and Network Analysis"

_ijms, 2025, doi:10.3390/ijms26125835_

Round 1

Reviewer 1 Report

Comments and Suggestions for Authors

The paper entitledAllosteric Coupling in Full-Length Lyn Kinase Revealed by Molecular Dynamics and Network Analysisby Mina Rabipour and co-workers characterizes the full-length Lyn kinase's regulatory dynamics and structural alterations upon binding ATP or Dasatinib. The paper provides a plethora of information concerning the above characteristics and can be helpful for future drug design. The paper is well done and can be accepted with some minor revisions, namely:

  1. The authors claim that Dasatinib binds to the ATP pocket through π-stacking and van der Waals interactions. Are these interactions identical to the experimental ones described in Williams’ article (J. Biol. Chem. 2009, 284, 284)?
  2. Are the representative interactions (l.185) hydrogen bonding or different intermolecular forces?
  3. A few recent reviews on the Lyn kinase are omitted from the paper, e.g., Bioorg. Med. Chem. Lett. 102 (2024)129674 (https://doi.org/10.1016/j.bmcl.2024.129674) or Frontiers in Immunology 2024 (DOI:10.3389/fimmu.2024.1395427)

Author Response

COMMENTS TO THE AUTHORS

"The paper entitled “Allosteric Coupling in Full-Length Lyn Kinase Revealed by Molecular Dynamics and Network Analysis” by Mina Rabipour and co-workers characterizes the full-length Lyn kinase's regulatory dynamics and structural alterations upon binding ATP or Dasatinib. The paper provides a plethora of information concerning the above characteristics and can be helpful for future drug design. The paper is well done and can be accepted with some minor revisions, namely:"

We thank Reviewer 1 for their insightful comments and constructive suggestions, which we have addressed in the revised manuscript (revised text is highlighted in red). Below are detailed responses to each point raised and corresponding revisions incorporated into the manuscript.

Comment 1: "The authors claim that Dasatinib binds to the ATP pocket through π-stacking and van der Waals interactions. Are these interactions identical to the experimental ones described in Williams’ article (J. Biol. Chem. 2009, 284, 284)?"

Response 1: We appreciate the Reviewer’s insightful query. Indeed, our MD simulations indicate that dasatinib engages residues in the hinge and β-sheet/αC helix region of the SH1 domain (e.g., Thr319, Met322, Ala323, Lys275, Met294), consistent with the experimental SH1 domain structure (PDB: 2ZVA) reported by Williams et al. (J. Biol. Chem. 2009). These similarities are explicitly described in the revised Results section (lines 196-220) and further discussed in the Discussion (lines 420-426), with a direct reference to the experimental study. Minor differences between MD and crystallographic observations are also addressed, reflecting the influence of full-length context and solvent exposure.

Comment 2: "Are the representative interactions (l.185) hydrogen bonding or different intermolecular forces?"

Response 2: Thank you for pointing out the need for clarification. As detailed in the revised Results (lines 201-206), the representative interactions primarily involve van der Waals forces and π-stacking, in line with dasatinib’s hydrophobic scaffold. However, transient hydrogen bonds with hinge backbone atoms were occasionally observed during the trajectory, and this nuance has now been explicitly stated in the manuscript.

Comment 3: "A few recent reviews on Lyn kinase are omitted from the paper, e.g., Bioorg. Med. Chem. Lett. 102 (2024) 129674 (https://doi.org/10.1016/j.bmcl.2024.129674) or Frontiers in Immunology 2024 (DOI:10.3389/fimmu.2024.1395427)."

Response 3: We thank the Reviewer for highlighting these recent and relevant reviews. In the revised manuscript, we have incorporated these references into the Introduction and Discussion sections, providing improved context for our findings within the latest developments in Lyn kinase research.

Reviewer 2 Report

Comments and Suggestions for Authors

This study provides a comprehensive investigation of the allosteric coupling mechanisms in full-length Lyn kinase across wild-type, ligand-bound, and cancer-associated mutant states using molecular dynamics simulations and network analysis. The research design is rigorous, the methodology is advanced, and the data are robust, offering significant insights into the dynamic regulation of Lyn kinase. I recommend acceptance after including the minor revisions.

  1. I am wondering the 16 interpretable features, but the supplementary video links are currently inaccessible. I would like to see the discusson of physical explaination of these features, and how its generalizability to other kinases.

  1. Figures 2 and 3 are well-annotated, but key residues in Figure 2b’s ΔRMSF plots could be labeled to clarify mutation-induced flexibility changes.

Author Response

COMMENTS TO THE AUTHORS

"This study provides a comprehensive investigation of the allosteric coupling mechanisms in full-length Lyn kinase across wild-type, ligand-bound, and cancer-associated mutant states using molecular dynamics simulations and network analysis. The research design is rigorous, the methodology is advanced, and the data are robust, offering significant insights into the dynamic regulation of Lyn kinase. I recommend acceptance after including the minor revisions."

We sincerely thank Reviewer 2 for their valuable comments, which have greatly contributed to the improvement of our manuscript. Below, we provide detailed responses and clearly outline the revisions made. All changes are incorporated in the revised manuscript and highlighted in red for clarity.

Comment 1: "I am wondering the 16 interpretable features, but the supplementary video links are currently inaccessible. I would like to see the discussion of physical explanation of these features, and how its generalizability to other kinases."

Response 1: We sincerely apologize for the inconvenience caused by the inaccessible supplementary video links. To rectify this, we have removed these links entirely. Instead, all supplementary materials, including the previously linked videos, have been provided as supplementary files compiled into a single zipped folder for easier accessibility.

Additionally, we have substantially expanded the Discussion section (lines 445-462) to include a physical interpretation of the 16 MD-derived features, describing their roles in ATP-binding pocket stability, interdomain coupling, and global structural transitions. We further discuss how many of these features (e.g., regulatory spine integrity, hinge salt-bridge networks) are conserved across Src-family kinases, supporting their generalizability to broader kinase systems. These revisions help clarify the mechanistic and transferable nature of our approach.

Comment 2: "Figures 2 and 3 are well-annotated, but key residues in Figure 2b’s ΔRMSF plots could be labeled to clarify mutation-induced flexibility changes."

Response 2: We thank the Reviewer for this valuable suggestion. In response, we have revised Figure 2b to label key residues that show significant ΔRMSF changes in response to mutations, particularly within SH1 and interdomain regions. We have also updated the figure legend of Figure 2b to reflect these annotations, providing clearer guidance to the reader. Corresponding text in the Results section (lines 135-141) now explains these residue-specific flexibility shifts and links them to functional implications of mutation-induced dynamics.

Reviewer 3 Report

Comments and Suggestions for Authors

The manuscript Allosteric Coupling in Full-Length Lyn Kinase Revealed by Molecular Dynamics and Network Analysis deals with dynamic and network-level framework for understanding Lyn regulation and identification of potential regulatory hotspots for structure-based drug design. The manuscript is well organized and written and certainly, it deserves to be published in this renamed journal. But, before publication, the author must resolve following issues:

  1. Taking into account wide readership of journal and different research areas, to clarify the importance of Lyn kinases, one paragraphs describing key physiological functions with corresponding literature sources should be added in Introduction section.
  2. The chemical structure of datasinib should be provided in article.
  3. One paragraph describing future directions in this field should be added at the end of article.

Author Response

COMMENTS TO THE AUTHORS

"The manuscript Allosteric Coupling in Full-Length Lyn Kinase Revealed by Molecular Dynamics and Network Analysis deals with dynamic and network-level framework for understanding Lyn regulation and identification of potential regulatory hotspots for structure-based drug design. The manuscript is well organized and written and certainly, it deserves to be published in this renamed journal. But, before publication, the author must resolve following issues:"

We thank Reviewer 3 for their positive evaluation of our manuscript and for highlighting its relevance and clarity. We appreciate the constructive suggestions to improve its accessibility and contextual relevance, particularly for a broader multidisciplinary readership. Below, we provide detailed responses to each point raised. All changes have been incorporated in the revised manuscript and are highlighted in red for clarity.

Comment 1: “Taking into account wide readership of journal and different research areas, to clarify the importance of Lyn kinases, one paragraph describing key physiological functions with corresponding literature sources should be added in Introduction section.”

Response 1: We thank the Reviewer and agree that providing context on the physiological relevance of Lyn kinase is important for non-specialist readers. Accordingly, we have added a paragraph in the Introduction (lines 28-40) summarizing Lyn’s functions in hematopoietic signaling, immune homeostasis, and oncogenesis. This paragraph also references recent literature, including studies highlighting the role of Lyn in autoimmune and neurodegenerative diseases.

Comment 2: “The chemical structure of dasatinib should be provided in article.”

Response 2: We appreciate this suggestion and agree with the Reviewer that including the chemical structure of dasatinib enhances clarity for readers. To maintain the flow of the main text while ensuring accessibility, we have included the chemical structure as Supplementary Figure S5. The structure was prepared using ChemDraw and includes the molecular formula (C₂₂H₂₆ClN₇O₂S). We reference this addition in the Methods section (subsection 4.2, lines 527-531), where dasatinib preparation is discussed.

Comment 3: “One paragraph describing future directions in this field should be added at the end of article.”

Response 3: We thank the Reviewer for this forward-looking suggestion. A dedicated paragraph has been added at the end of the Discussion section (lines 496-508). It outlines emerging directions in computational kinase drug discovery, such as structure-guided mutagenesis, experimental validation of predicted allosteric sites using biosensors, and integration of AI with MD simulations for rational drug design. These strategies aim to advance selective allosteric targeting in Lyn and other kinases.